# BMJ Open  Is interpregnancy interval associated with cardiovascular risk factors in later life? A cohort study

Duleeka W Knipe,[1] Abigail Fraser,[1,2] Debbie A Lawlor,[1,2] Laura D Howe[1,2]

▶ Additional material is available. To view please visit the journal (http://dx.doi.org/10.1136/bmjopen-2013-004173).

This work was carried out at the School of Social and Community Medicine, University of Bristol

[1]School of Social and Community Medicine, University of Bristol, Bristol, UK
[2]MRC Integrative Epidemiology Unit at the University of Bristol, Bristol, UK

**Correspondence to**
Duleeka W Knipe;
dee.knipe@bristol.ac.uk

## ABSTRACT

**Objectives:** Pregnancy represents a metabolic challenge to women; in a normal pregnancy, transient metabolic changes occur that support the needs of the growing fetus. It is possible that repeating this challenge within a relatively short amount of time may result in lasting damage to the woman's cardiovascular health. Conversely, it is also possible that a long interpregnancy interval (IPI) may reflect subfertility, which has been found to be associated with cardiovascular disease (CVD). We examine the associations of short and long IPI with measures of cardiovascular health.

**Design:** Prospective cohort.

**Setting:** Mothers of the Avon Longitudinal Study of Parents and Children (ALSPAC).

**Participants:** Women with two live births in order to control for confounding by parity.

**Outcome measures:** Arterial distensibility, common carotid intima, adiposity, blood pressure, lipids, glucose, insulin, proinsulin, triglycerides, C reactive protein.

**Results:** 25% (n=3451) of ALSPAC mothers had provided sufficient data to determine full reproductive history—of these, 1477 had two live births, with 54% mothers having non-missing data on all variables required for our analyses. A total of 1268 mothers with IPI (interbirth interval minus 9 months' gestation) had CVD risk factors measured/imputed at mean age 48 years. After adjusting for confounding, we found no association of either short (≤15 months) or long (>27 months) IPI and increased levels of cardiovascular risk factors. There was some suggestion that women with long and short IPIs had a more favourable lipid profile compared with women whose IPI was 16–27 months; however, the differences were small in magnitude and imprecisely estimated.

**Conclusions:** This study does not support the hypothesis that either long or short IPI is a risk factor for later cardiovascular health.

### Strengths and limitations of this study

- Its prospective design with detailed data on reproductive history.
- The availability of a wide range of objectively measured cardiovascular disease risk factors and the large sample size.
- First study to examine the association of interpregnancy interval (IPI) with cardiovascular risk factors.
- As with other prospective cohort studies, there was loss to follow-up, with those attending and completing all questionnaires tending to come from a higher socioeconomic background.
- Our analysis was restricted to women who had two live births to remove confounding by parity, which means that our findings may not generalise to women who had three or more children. We did not find any evidence of any strong differences between women who had either only one birth or three or more births and women with two live births/stillbirths.
- IPI was calculated by subtracting the average gestation period from the birth interval. We do not believe that this would have biased our findings as we used IPI as a categorical variable, and therefore any fluctuations around the average 9-month gestation would not have altered the findings substantially.
- Only generalisable to a largely white European population.
- The population being studied is still young (mean age at clinic=48), and therefore it is possible that the association may emerge at older ages, when interindividual variability in cardiovascular risk factors becomes greater.

## INTRODUCTION

Pregnancy represents a metabolic challenge to women; in a normal pregnancy, transient metabolic changes occur that support the needs of the growing fetus. Women become relatively insulin resistant, hyperlipidaemic and experience upregulation of coagulation factors and the inflammatory cascade.[1] A recent systematic review and meta-analysis found that a short interpregnancy interval (IPI; <12 months) was associated with an increased risk of stillbirths, early neonatal death, preterm birth and low birth weight.[2] Associations between IPI and cardiovascular outcomes, however, are not known. Given

the cardiovascular changes during pregnancy, it is possible that if this challenge is repeated within a relatively short amount of time, the effects on a woman's metabolic system may be exacerbated and/or be longer lasting. This may have a deleterious effect on her long-term cardiovascular health.

Conversely, it is also possible that a long IPI is associated with increased cardiovascular risk. Longer IPIs may reflect subfertility, which has been found to be associated with cardiovascular disease (CVD).[3] [4]

Here we examine the associations of short and long IPI with measures of cardiovascular health (arterial distensibility, common carotid intima, adiposity, blood pressure, lipids, glucose, insulin, proinsulin, triglycerides, C reactive protein (CRP)) assessed at a mean age of 48 years.

## METHODS

The Avon Longitudinal Study of Parents and Children (ALSPAC) is a prospective population-based birth cohort that recruited 14 541 pregnant women resident in Avon, UK, who had expected delivery dates between 1 April 1991 and 31 December 1992. ALSPAC has previously been described in detail,[5] and the study website contains details of all the available data through a fully searchable data dictionary (http://www.bris.ac.uk/alspac/researchers/data-access/data-dictionary).

### Pregnancy data and IPI

In order to remove confounding by parity, which is known to be associated with CVD,[6–12] we restrict our analyses to women with two pregnancies resulting in live births (self-reported). Women were eligible for inclusion in this study if they had two pregnancies that resulted in live births and completed all the questionnaires necessary to ascertain full reproductive history. The study flow diagram is presented in figure 1.

On recruitment into the cohort (approximately 18 weeks' gestation), women were asked to complete a questionnaire about their health prior to the current pregnancy, including the number of previous pregnancies; the number of miscarriages/abortions; the outcome (live birth, stillbirth, miscarriage, abortion or termination, live born baby that died, live born baby still alive and other) and end date (ie, delivery or other outcome) of their most recent previous pregnancy. We used these data, along with subsequent questionnaires (n=5) throughout the 18-year period, leading to our outcome assessment to identify women who had two pregnancies that resulted in a live birth. We only included pregnancies that resulted in a live birth and the two pregnancies that we used for the interval could have included those occurring before or after the pregnancy at which women were recruited.

### Assessment of cardiovascular risk factors

Between 2009 and 2011, ALSPAC mothers (N=11, 264 women) were invited to a research clinic assessment at which a range of cardiovascular outcomes were assessed;

this clinic took place between 1.6 and 20.3 years (median 18) since the second birth defining the end of the IPI. Not all of these women were eligible for inclusion in our analysis as not all of them had full reproductive histories recorded.

Carotid intima media thickness (cIMT) for the left and right common carotid artery scans were obtained via high-resolution B ultrasound and imaged longitudinally 1 cm proximal to the carotid bifurcation following a standardised protocol using a ZONARE z.one Ultra convertible ultrasound system with an L10-5 linear transducer. Images were focused on the posterior (far) wall of the artery and the zoom function was used to magnify the area. Ten-s cine loops were recorded in DICOM format and analysed offline using Carotid Analyser for Research (Vascular Research Tools 5, Medical Imaging Applications, LLC 2008). Three consecutive cardiac cycles were identified and three measures of cIMT were taken from end-diastolic frames and averaged. This was carried out for right and left carotid arteries. Arterial distensibility was calculated as the difference between systolic and diastolic arterial diameter. The mean of the left-sided and right-sided readings was used in analyses. The images were analysed by a single trained reader. Blood pressure was measured while the women were lying down with the use of an Omron M6 monitor (Omron Healthcare UK Ltd, Milton Keynes, UK). Two readings of systolic and diastolic blood pressure were recorded on each arm, and the mean of these four readings was used here. Heart rate was measured in the sitting and standing positions.

Blood samples were taken after an overnight fast for those attending in the morning or after a minimum 6 h fast for those attending the clinic after 14:00. Blood samples were obtained, centrifuged, separated and frozen at −80°C within 30 min. Plasma glucose was measured by the automated enzymatic (hexokinase) method. Plasma insulin was measured by ELISA (Mercodia, Uppsala, Sweden) that does not cross react with proinsulin or c-peptide, and proinsulin was also measured by ELISA (Mercodia) that is a solid-phase 2-site enzyme immunoassay for the quantification of human proinsulin. Lipids were measured by automated analyser with enzymatic methods. CRP was measured by automated particle-enhanced immunoturbidimetric assay (Roche UK, Welwyn Garden City, UK).

Whole body fat mass was measured using dual-energy X-ray absorptiometry whole body scans. Weight and height were measured with the participants in light clothing and without shoes. Weight was measured to the nearest 0.1 kg with the use of Tanita scales. Height was measured to the nearest 0.1 cm with a Harpenden stadiometer. Waist circumference was measured twice to the nearest 1 mm at the midpoint between the lower ribs and the pelvic bone with a flexible tape. The mean of the two measures is used here.

A total of 4834 women attended the clinic (43% response).[5] Women attending the clinic had a mean age

**Figure 1** Flow chart of selection of women for analysis. *All 5 questionnaires were not completed.

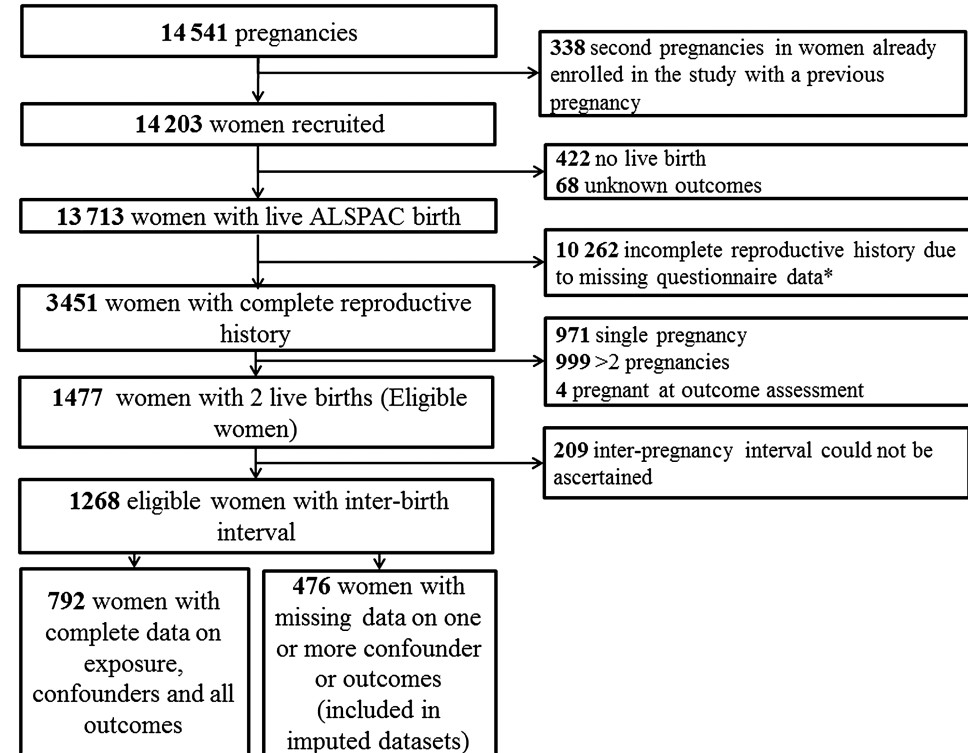

of 48 years. Women pregnant at the time of clinic assessment were excluded from our analyses (n=7). Only 954 women attending the clinic and not pregnant at the time of clinic assessment had full reproductive histories.

## Measurement of confounding factors

Information on pre-pregnancy weight and height, smoking in pregnancy, single parent households, ethnicity, education and social class were obtained from questionnaires completed at the time of recruitment during the index pregnancy. Ethnicity was categorised as white or non-white. Education was categorised as below or above university education. Household occupational social class was defined according to the 1991 British Office of Population and Census Statistics classification (classes I (professional/managerial) to V (unskilled manual workers)) using either the woman's occupation or that of her partners, whichever was highest. Smoking in pregnancy was categorised as ever or never smoked during pregnancy, at approximately 18–20 weeks' gestation. Ideally, we would wish to measure these potential confounders in relation to the first pregnancy of the interval and for some to have time updated (ie, also information from the second pregnancy) measurements. In this study, however, we only have data on the pregnancy when women were recruited, and this is the first pregnancy of the interval for 30.9% of the women and the second for the remaining 69%. Participant's age at clinic attendance was included as a confounding factor.

## Statistical analysis

Distributions of insulin, proinsulin, triglyceride, CRP, glucose and low-density lipoprotein were right skewed, and so were log transformed for all regression model analyses, which ensured that the residuals were approximately normally distributed. We compared women with sufficient data for us to detail their full reproductive history with those women with insufficient data, in terms of all confounding variables. Similarly, we compared those women who formed our eligible sample (only two live births and not pregnant at the time of outcome assessment) with women with either only 1 birth or ≥3 births, in terms of all confounding and outcome variables. $\chi^2$ Tests for categorical data and t tests for continuous normally distributed variables were performed. IPI was calculated as the difference between the dates of delivery of the two pregnancies, minus the average gestation period (9 months). IPI was grouped into three categories: ≤15, 16–27 and >27 months; these groups correspond to ≤2, 2–3 and >3 years between births.

The association between our categorised measure of IPI and each outcome was assessed using multiple linear regression, with and without adjustment for potential confounders, comparing both short and long IPI with the reference category of 16–27 months. All analysis was conducted in Stata/MP V.12.0.

In order to assess whether our results were biased as a result of some confounders being measured at the first pregnancy of the IPI for some women and the second for others, we stratified our results by whether the ALSPAC child was the first or second pregnancy of the interval.

## Dealing with missing data

Within our eligible study sample of 1268, there were some missing data on potential confounders and outcome measurements (data on exposure had to be observed to be

eligible). The extent of missing data varied from 0% to 29%, with blood-based measures having the most missing data (see online supplementary table S1). In order to increase efficiency and minimise selection bias, we imputed missing variables for eligible participants who had missing data on outcomes or confounders, using multivariate multiple imputation. We included all the exposure, confounder and outcome measures in the imputation equations. We used switching regression in Stata and carried out 20 cycles of regression switching and generated 20 imputation datasets. The main analysis results are obtained by averaging across the results from each of these 20 datasets using Rubin's rules and the SEs for any regression coefficients (used to calculate p values and 95% CIs) to take account of uncertainty in the imputations as well as uncertainty in the estimate.[13] We carried out all our linear regression analyses using the imputed and complete-case datasets. Online supplementary table S1 shows that the distributions of variables in the imputed and non-imputed/complete-case datasets were similar.

## RESULTS
### Sample description
A total of 3451 women of 13 713 recruited women who had a live birth had sufficient data to describe a full reproductive history; of these, 1970 were excluded because they had either just 1 (n=971) or ≥3 births (n=999). A further four were pregnant at the time of assessment and were excluded. Of the remaining 1477 women who had two live births, 147 were excluded as they were recruited into the cohort during their second pregnancy, with the most recent previous pregnancy resulting in a miscarriage, termination,

stillbirth or 'other' outcome. IPI could be calculated for 1268 of the remaining eligible women (95%). Of these women, 954 attended the clinic assessment in 2009–2011 and 792 women had complete data for all outcome and confounder measures (figure 1). Thus, our complete case analysis was performed on 792 women, and analysis using multivariate multiple imputation included 1268 women. Women included in the complete case analysis versus women with missing data on covariates and/or outcome variables had similar IPI (see online supplementary table S2).

Women who had provided sufficient data to describe their full reproductive history were on average slightly older, from a higher socioeconomic background, more likely to be smokers and white, and less likely to be in a single parent household, compared with women with insufficient data to describe their full reproductive history (table 1). Women eligible for this study (two live births) were more likely to be smokers compared to women with either 1 or ≥3 births (see online supplementary table S3). No clear differences with regard to the other confounding variables were observed. There were small differences observed in terms of cardiovascular risk factors (see online supplementary table S3), with women with single and three or more pregnancies having slightly worse cardiovascular risk profile compared to women with two pregnancies.

### Participant characteristics
The characteristics of women according to their IPI are presented in table 2. There were no noticeable differences between IPI categories in terms of pre-pregnancy body mass index (BMI), ethnicity, social class, education and

**Table 1** Characteristics of women with full reproductive histories compared to those with incomplete reproductive histories as determined from five questionnaires

| | Women with incomplete data n=10262 | Women with complete data n=3451 | p Value |
|---|---|---|---|
| Categorical variables—n (%) | | | |
| Ethnicity | n=8653 | n=3417 | |
| White | 8381 (96.9) | 3370 (98.6) | <0.001 |
| Social class | n=7927 | n=3335 | |
| I | 842 (10.6) | 664 (19.9) | <0.001 |
| II | 3099 (39.1) | 1601 (48.0) | |
| III (non-manual) | 2135 (26.9) | 738 (22.1) | |
| III (manual) | 1261 (15.9) | 258 (7.7) | |
| IV and V | 590 (7.4) | 74 (2.2) | |
| Education | n=8739 | n=3425 | |
| University level or above | 826 (9.5) | 742 (21.7) | <0.001 |
| Single parent household | n=9136 | n=3403 | |
| Yes | 616 (6.7) | 102 (3.0) | <0.001 |
| Ever smoked | n=9365 | n=3441 | |
| Yes | 5124 (54.7) | 1369 (39.8) | <0.001 |
| Continuous variables—mean (SD*) | | | |
| Age at birth of the ALSPAC | n=10258 | n=3451 | |
| index child | 27.4 (5.0) | 29.8 (4.4) | <0.001 |

*T test for normally distributed data.
ALSPAC, Avon Longitudinal Study of Parents and Children.

**Table 2** Characteristics of women by interpregnancy interval N=1268

| | Interpregnancy interval of all included women n=1268 | | | |
| --- | --- | --- | --- | --- |
| | ≤15 months n=269 | 16–27 months n=412 | >27 months n=582 | p Value |
| Continuous variables—mean (SD) | | | | |
| Age at birth of the ALSPAC index child | n=269 | n=412 | n=582 | |
| | 30.4 (3.7) | 30.2 (3.6) | 29.3 (4.3) | <0.001 |
| Pre-pregnancy BMI (kg/m$^2$)* | n=256 | n=391 | n=534 | |
| | 22.6 (2.9) | 22.5 (3.4) | 22.7 (3.4) | 0.6 |
| Categorical variables—n (%) | | | | |
| Ethnicity | n=267 | n=411 | n=577 | |
| White | 265 (99.3) | 406 (98.8) | 566 (98.1) | 0.4 |
| Social class | n=261 | n=409 | n=568 | |
| I (highest; professional) | 59 (22.6) | 89 (21.8) | 109 (19.2) | |
| II | 127 (48.7) | 198 (48.4) | 274 (48.2) | |
| III (non-manual) | 54 (20.7) | 74 (18.1) | 141 (24.8) | |
| III (manual) | 17 (6.5) | 37 (9.1) | 34 (6.0) | |
| IV and V (lowest; unskilled manual) | 4 (1.5) | 11 (2.7) | 10 (1.8) | 0.2 |
| Education | n=269 | n=411 | n=580 | |
| University level or above | 66 (24.5) | 82 (20.0) | 124 (21.4) | 0.4 |
| Single parent household | n=269 | n=412 | n=578 | |
| Yes | 1 (0.4) | 3 (0.7) | 24 (4.2) | <0.001 |
| Smoked during pregnancy* | n=268 | n=412 | n=581 | |
| Yes | 101 (38.0) | 152 (37.0) | 2013 (36.7) | 1.0 |

*Relates to the first child in some women and second in others.
ALSPAC, Avon Longitudinal Study of Parents and Children; BMI, body mass index.

smoking status. Women with a longer IPI, however, were more likely to be in single parent households and slightly younger compared with all other women (table 2). Table 3 presents the distribution of the outcome variables for all those women who attended the follow-up assessment in 2009–2011 and who had an IPI calculated for this analysis.

## Association between IPI and CVD risk factors

Crude associations of IPI with outcomes are presented in tables 4 and 5. We found no evidence that women with either short or long IPI had more adverse cardiovascular risk factors compared with women with an IPI between 16 and 27 months. There was some suggestion

**Table 3** Distribution of pre-pregnancy BMI and outcomes at research clinic assessment (2009–2011) with IPI

| | | IPI n=1268 | | | | | |
| --- | --- | --- | --- | --- | --- | --- | --- |
| | n= | ≤15 months n=271 | n= | 16–27 months n=415 | n= | >27 months n=582 | |
| Mean (SD) | | | | | | | |
| Pre-pregnancy BMI (kg/m$^2$) | 260 | 22.6 (2.8) | 399 | 22.5 (3.4) | 553 | 22.7 (3.4) | |
| BMI (kg/m$^2$) | 203 | 25.9 (4.5) | 313 | 25.8 (4.7) | 433 | 26.1 (4.8) | |
| Waist (cm) | 203 | 82.6 (11.4) | 313 | 82.6 (11.2) | 433 | 83.3 (11.2) | |
| SBP (mm Hg) | 202 | 118.1 (12.5) | 304 | 118 (12.3) | 424 | 117.4 (12.5) | |
| DBP (mm Hg) | 202 | 71.7 (8.3) | 304 | 71.7 (8.5) | 424 | 71.4 (8.1) | |
| Heart rate | 202 | 84.3 (11.1) | 304 | 83.7 (10.7) | 424 | 83.6 (11) | |
| Fat mass (kg) | 204 | 26 (10.1) | 314 | 25.9 (9.9) | 428 | 26.3 (10.1) | |
| Average arterial distensibility (mm) | 204 | 0.5 (0.1) | 315 | 0.5 (0.1) | 430 | 0.5 (0.1) | |
| Common carotid intima (mm) | 204 | 0.6 (0.1) | 315 | 0.6 (0.1) | 430 | 0.6 (0.1) | |
| Cholesterol (mmol/L) | 194 | 4.9 (0.8) | 303 | 5 (0.9) | 410 | 4.9 (0.9) | |
| HDL (mmol/L) | 194 | 1.5 (0.4) | 303 | 1.5 (0.4) | 410 | 1.5 (0.4) | |
| Non-HDL cholesterol (mmol/L) | 194 | 3.4 (0.9) | 303 | 3.5 (0.9) | 410 | 3.4 (0.9) | |
| Median (IQR) | | | | | | | |
| Insulin (μ/L) | 193 | 4.3 (3.2 to 6.4) | 301 | 4.3 (3 to 6.3) | 408 | 4.6 (3.3 to 6.8) | |
| Proinsulin (pu/L) | 194 | 5 (3.6 to 7.3) | 302 | 4.9 (3.5 to 6.6) | 409 | 5.1 (3.6 to 7.9) | |
| Triglyceride (mmol/L) | 194 | 0.8 (0.6 to 1.1) | 303 | 0.9 (0.7 to 1.2) | 410 | 0.9 (0.7 to 1.1) | |
| C reactive protein (mg/L) | 194 | 0.9 (0.4 to 1.9) | 303 | 0.9 (0.5 to 2) | 410 | 1 (0.5 to 2.5) | |
| Glucose (mmol/L) | 194 | 5.2 (4.9 to 5.4) | 303 | 5.2 (4.9 to 5.4) | 410 | 5.1 (4.9 to 5.4) | |

BMI, body mass index; DBP, diastolic blood pressure; HDL, high-density lipoprotein; IPI, interpregnancy interval; SBP, systolic blood pressure.

that women with long and short IPIs had a more favourable lipid profile compared with women whose IPI was 16–27 months; however, the differences were small in magnitude and imprecisely estimated, with the 95% CI including the null value.

Adjustment for potential confounders or for potential mediation by BMI at outcome assessment did not change results (tables 4 and 5 and online supplementary table S4). These findings are similar to the results when the analysis was restricted to complete cases (see online supplementary tables S5 and S6). The results were also similar when the women were stratified according to whether the ALSPAC child was the first or second pregnancy (results not shown—available from authors on request).

## DISCUSSION

We found no association between short or long IPI and levels of cardiovascular risk factors in a cohort of women with a mean age of 48 years. This is in contrast to our initial hypotheses that shorter IPIs may give a woman's cardiovascular system insufficient time to recover, that longer IPIs may reflect subfertility and that both may be associated with adverse levels of cardiovascular risk factors. We actually found weak evidence that women with short (≤15 months) and long IPIs (>27) had a more favourable lipid profile (ie, the opposite of our initial hypotheses) compared with women with an IPI between 16 and 27 months, but the associations were

not large, were imprecisely estimated, and were not consistent across all outcomes and may be due to chance.

To the best of our knowledge, this is the first study to examine the relationship between pregnancy interval and subsequent cardiovascular outcomes. Several previous studies have examined the association of parity with cardiovascular outcomes, with most,[6–12] though not all,[14–16] finding that greater parity is related to more adverse risk factors and greater disease risk. A recent study investigated this association in a population of 1.3 million, using the Swedish registry data,[12] and found a J-shaped relationship between parity and CVD risk, with nulliparous and grand multiparous (≥5 births) having elevated risk compared to those women with two births. One possible mechanism for this association is that multiparous women are more likely to have births closer together, and the repetition of the cardiovascular challenge within a relatively short amount of time may lead to the effects on a women's metabolic system being exacerbated and/or longer lasting. Our findings do not support this hypothesis. Our results therefore suggest that the association of parity with greater CVD risk may be through another mechanism. Possible theories for the mechanisms underlying the greater risk of CVD after two children are: (1) other adverse lifestyle factors being adopted as a family size increases; (2) sociodemographic and other characteristics associated with increased risk of CVD also being associated with having more children and (3) adverse metabolic disturbances accumulating over pregnancies.[12]

**Table 4** Unadjusted and confounder-adjusted analysis of the association between interpregnancy interval and blood-based cardiovascular risk factors using multivariate multiply imputed data (N=1268)

| | Analysis model number | Interpregnancy interval | | | | |
| | | ≤15 months | | 16–27 months | >27 months | |
| | | β* (95% CI) | p Value | Ref group | β* (95% CI) | p Value |
|---|---|---|---|---|---|---|
| Cholesterol (mmol/L) | Model 1† | −0.1 (−0.2 to 0.1) | 0.3 | 0 | −0.1 (−0.2 to 0.03) | 0.1 |
| | Model 2‡ | −0.1 (−0.3 to 0.1) | 0.2 | 0 | −0.05 (−0.2 to 0.1) | 0.4 |
| HDL (mmol/L) | Model 1† | 0.03 (−0.04 to 0.1) | 0.4 | 0 | −0.01 (−0.1 to 0.05) | 0.8 |
| | Model 2‡ | 0.03 (−0.04 to 0.1) | 0.4 | 0 | 0.01 (0 to 0.1) | 0.6 |
| | | Per cent change (95% CI) | p Value | Ref group | Per cent change (95% CI) | p Value |
| Insulin (μ/L)§ | Model 1† | 1 (0.9 to 1.1) | 0.8 | 0 | 1.1 (1 to 1.2) | 0.1 |
| | Model 2‡ | 1 (0.9 to 1.1) | 0.9 | 0 | 1 (1 to 1.1) | 0.4 |
| Proinsulin (pmol/L)§ | Model 1† | 1 (0.9 to 1.1) | 0.7 | 0 | 1.1 (1 to 1.1) | 0.1 |
| | Model 2‡ | 1 (0.9 to 1.1) | 0.8 | 0 | 1 (1 to 1.1) | 0.2 |
| Triglyceride (mmol/L)§ | Model 1† | 0.9 (0.9 to 1) | 0.1 | 0 | 0.9 (0.9 to 1) | 0.03 |
| | Model 2‡ | 0.9 (0.9 to 1) | 0.1 | 0 | 0.9 (0.9 to 1) | 0.02 |
| CRP (mg/L)§ | Model 1† | 0.9 (0.8 to 1.1) | 0.4 | 0 | 1.1 (0.9 to 1.2) | 0.4 |
| | Model 2‡ | 0.9 (0.8 to 1.1) | 0.5 | 0 | 1 (0.9 to 1.2) | 0.6 |
| Glucose (mmol/L) § | Model 1† | 1 (1 to 1) | 0.7 | 0 | 1 (1 to 1) | 0.4 |
| | Model 2‡ | 1 (1 to 1) | 0.6 | 0 | 1 (1 to 1) | 0.3 |

*Mean difference compared with 16–27 months.
†Unadjusted model.
‡Adjusted for maternal age, maternal ethnicity, social class, education, pre-pregnancy BMI, ever smoking during pregnancy.
§Log transformed.
BMI, body mass index; CRP, C reactive protein; HDL, high-density lipoprotein.

**Table 5** Unadjusted and confounder adjusted analysis of the association between interpregnancy interval and non-blood-based cardiovascular risk factors using multivariate multiply imputed data (N=1268)

| | | Interpregnancy interval | | | | | |
| | Analysis model number | ≤15 months | | | 16–27 months Ref group | >27 months | |
| | | β* (95% CI) | p Value | | | β* (95% CI) | p Value |
|---|---|---|---|---|---|---|---|
| Average arterial distensibility (mm) | Model 1† | −0.0003 (−0.02 to 0.02) | 1 | | 0 | 0.02 (0.001 to 0.04) | 0.04 |
| | Model 2‡ | 0.002 (−0.02 to 0.02) | 0.9 | | 0 | 0.01 (−0.003 to 0.03) | 0.1 |
| Common carotid intima (mm) | Model 1† | 0.005 (−0.01 to 0.02) | 0.4 | | 0 | −0.0008 (−0.01 to 0.01) | 0.9 |
| | Model 2‡ | 0.003 (−0.01 to 0.01) | 0.6 | | 0 | 0.002 (−0.01 to 0.01) | 0.7 |
| Total body fat mass (kg)§ | Model 1† | −0.2 (−1.7 to 1.3) | 0.8 | | 0 | 0.4 (−0.8 to 1.7) | 0.5 |
| | Model 2‡ | −0.3 (−1.6 to 0.9) | 0.6 | | 0 | −0.05 (−1.1 to 1) | 0.9 |
| BMI (kg/m²) | Model 1† | 0.03 (−0.7 to 0.8) | 0.9 | | 0 | 0.3 (−0.3 to 0.9) | 0.3 |
| | Model 2‡ | −0.02 (−0.6 to 0.5) | 0.9 | | 0 | 0 (−0.4 to 0.5) | 0.8 |
| Waist (cm) | Model 1† | −0.1 (−1.8 to 1.6) | 0.9 | | 0 | 0.6 (−0.8 to 2.1) | 0.4 |
| | Model 2‡ | −0.3 (−1.7 to 1.1) | 0.7 | | 0 | 0.1 (−1.1 to 1.4) | 0.8 |
| Systolic BP (mm Hg) | Model 1† | 0.1 (−1.9 to 2.2) | 0.9 | | 0 | −0.6 (−2.3 to 1.2) | 0.5 |
| | Model 2‡ | 0.1 (−1.9 to 2.1) | 0.9 | | 0 | −0.5 (−2.3 to 1.2) | 0.5 |
| Diastolic BP (mm Hg) | Model 1† | 0.1 (−1.3 to 1.4) | 0.9 | | 0 | −0.4 (−1.6 to 0.8) | 0.5 |
| | Model 2‡ | 0.1 (−1.2 to 1.5) | 0.9 | | 0 | −0.5 (−1.7 to 0.7) | 0.4 |
| Heart rate | Model 1† | 0.4 (−1.6 to 2.4) | 0.7 | | 0 | 0 (−1.7 to 1.6) | 1 |
| | Model 2‡ | 0.3 (−1.7 to 2.4) | 0.7 | | 0 | 0 (−1.6 to 1.7) | 1 |

*Mean difference compared with 16–27 months.
†Unadjusted model.
‡Adjusted for age, ethnicity, social class, education, pre-pregnancy BMI, ever smoking during pregnancy.
§Height and height² were included into the model.
BP, blood pressure; BMI, body mass index.

The strengths of this study include the prospective design; detailed data on reproductive history; the availability of a wide range of objectively measured CVD risk factors; and the large sample size. We were able to adjust for a range of potential confounding factors, and we restricted our analyses to women who had two live births in order to remove confounding by parity. To the best of our knowledge, this is the first study to examine the association of IPI with cardiovascular risk factors. The findings of this study, however, should be considered in light of several limitations. As with other prospective cohort studies, there was loss to follow-up, with those attending and completing all questionnaires tending to come from a higher socioeconomic background. This means that the women included in our analysis are not a representative sample of the full ALSPAC cohort; some studies suggest that lack of generalisability does not necessarily result in selection bias,[17–20] but we cannot be certain of this from the current analysis, and replication of our results in other studies with different distributions of sociodemographic variables would be beneficial. Missing data would bias our results if the association between IPI and cardiovascular risk factors differed in those included in our analyses and those excluded due to missing data. We are unable to test this assumption, but have no reason to suspect that it may be violated. One important consideration when interpreting our results is that we have restricted our analyses to women who had two live births. While we feel that this was a sensible analysis

strategy in order to remove confounding by parity, it means that our findings may not generalise to women who had three or more children. We did not find any evidence of any strong differences between women who had either only one birth or three or more births and women with two live/stillbirths (see online supplementary table S3). IPI was calculated by subtracting the average gestation period (9 months) from the birth interval; ideally, we would have liked to have calculated exact gestation periods for each live birth. We do not, however, believe that this would have biased our findings as we used IPI as a categorical variable, and therefore any fluctuations around the average 9 month gestation would not have altered the findings substantially. It is possible that by calculating our IPI in this way we have attenuated our results towards the null. A further limitation of this study is these findings are only generalisable to a largely white European population with a higher socioeconomic status. As the population being studied is still young (mean age at clinic=48), it is possible that the association may emerge at older ages, when interindividual variability in cardiovascular risk factors becomes greater. Ideally, we would have measured pre-pregnancy BMI and smoking during pregnancy for the first pregnancy of all women. Owing to our study design, this was not possible. These measurements are from the first pregnancy in some women and the second pregnancy in others. There is therefore the possibility that these measurements are not a reasonable representation of levels

in the first pregnancy for all women; this may lead to residual confounding. However, given that the associations we observe are null, we do not think this has biased our results.

In conclusion, our results do not support an association between IPI and cardiovascular risk factors, though our findings must be interpreted in light of the large losses to follow-up and limitations in our measurement of IPI. Further studies in other populations with more detailed data on gestational age at delivery of all pregnancies, in different settings such as low-income countries where the social patterning of IPI may differ and in older women with greater interindividual variability in cardiovascular risk factors would provide a more comprehensive understanding of these associations.

**Acknowledgements** The authors are extremely grateful to all the families who took part in this study, the midwives for their help in recruiting them, and the whole Avon Longitudinal Study of Parents and Children (ALSPAC) team, which includes interviewers, computer and laboratory technicians, clerical workers, research scientists, volunteers, managers, receptionists and nurses. This publication is the work of the authors and DWK will serve as the guarantor for the contents of this paper.

**Contributors** DAL, LDH and AF conceived the study idea; DWK, LDH and AF developed the analysis plan; DWK undertook the analysis under the supervision of LDH and AF; DWK wrote the first draft of the paper and collated coauthor feedback; and all authors contributed to the critical review and final version.

**Funding** The work was supported by the British Heart Foundation (SP/07/008/24066), the UK Medical Research Council (0701594 to AF and G1002375 to LDH) and the Wellcome Trust (and WT099874MA to DWK). AF, DAL and LDH work in a unit that receives funding from the UK Medical Research Council and the University of Bristol. The UK Medical Research Council ((074882), the Wellcome Trust (WT076467) and the University of Bristol provide core funding support for Avon Longitudinal Study of Parents and Children (ALSPAC).

**Competing interests** None.

**Ethics approval** Avon Longitudinal Study of Parents and Children (ALSPAC) Ethics and Law Committee and the Local Research Ethics Committees.

**Provenance and peer review** Not commissioned; externally peer reviewed.

**Data sharing statement** This study is based on the Avon Longitudinal Study of Parents and Children (ALSPAC) study. ALSPAC has a detailed data sharing policy which can be found at: http://www.bristol.ac.uk/alspac/researchers/data-access/policy/

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
