## [Reviewer comments · BMJ Open]

Some articles will have been accepted based in part or entirely on reviews undertaken for other BMJ Group journals. These will be reproduced where possible.

ARTICLE DETAILS

TITLE (PROVISIONAL)	Is inter-pregnancy interval associated with cardiovascular risk factors in later life? : a cohort study
AUTHORS	Knipe, Duleeka; Fraser, Abigail; Lawlor, Debbie; Howe, Laura

VERSION 1 - REVIEW

REVIEWER	Janet Catov University of Pittsburgh, USA
REVIEW RETURNED	29-Oct-2013

GENERAL COMMENTS	This is a nicely written paper that studies an interesting question, whether a short or long inter-pregnancy interval may be related to maternal CVD risk. The study utilizes a large prospective cohort with CVD assessment on average 20 years after the studied pregnancy. Their hypothesis is that a short or long interval may both be related to excess CVD risk and they have a comprehensive set of biomarkers and vascular outcome measures. The study is well designed, and the authors make a strong case for limiting their analysis to women with only two births to eliminate confounding by parity. It could be argued that a preferable design might be to include all women with 2 or more births, adjust for parity and provide a metric for assessing the interval length when assessed across more than 2 births. However, it is not clear that an alternative approach would result in a different answer. And yet, as this cohort is one of the few with lengthy follow up it can help set the standard for how to study reproductive events that accumulate across a life course. The authors do an adequate job, however, of explaining their rationale. Although the findings are null, the study makes an important contribution, its methods are robust and therefore the paper is important. A few questions and comments: 1. It is not clear at what time age is reported for Table 1. I had thought it was age at the index ALSPAC birth, but that does not seem to be the case given the ages provided in Table 2.2. It is not clear why the mean values for the entire measured cohort (~4000) are reported in table 3 when they are not compared to the subcohort analyzed. It perhaps would have been more informative to see the mean values summarized by interval timing, as for table 23. Tables 4 & 5 are a bit hard to follow. IBI is not defined. It appears that the models noted as 'a' are unadjusted, and 'b' are adjusted; but then I am not certain what the numbers refer to (1a, 2b)
--

REVIEWER	Cassandra Gibbs Pickens, MPH Emory University, Rollins School of Public Health, Department of Epidemiology
REVIEW RETURNED	16-Dec-2013

GENERAL COMMENTS

I will address my 'no' responses below. They are numbered according to the question that they address. I use IPI as an abbreviation for interpregnancy interval.

2 (abstract)-You need to mention attrition/missing data in the abstract since it is such a large limitation of this study. You should mention the % of women in the ALSPAC Study with full reproductive history (approximately 25%). You should also mention the % of women with full reproductive history (& who are eligible due to having exactly 2 live births) who have non-missing data (approximately 56%). You should also mention how you determined interpregnancy interval (interbirth interval minus the average GA, 9 months) in the abstract.

7 (statistics)- Three major comments. 1. What about control for breastfeeding duration? Breastfeeding may affect IPI (through lactation-induced amenorrhea) and may also decrease the risk of adverse CVD outcomes later in life and could be an important confounder to include in your adjusted models. I suppose you only have information on breastfeeding for women who filled out questionnaires for their second pregnancy (not their first), but it would be a good confounder to include if you ran a model among these women only. If you don't have this information, listing it as a potential limitation would be a good idea.

2. Why do you include BMI at follow-up as a mediating variable for the IPI CVD risk factor association? Controlling for maternal BMI at follow-up indicates that some of the IPI CVD risk factor association is mediated through maternal BMI at follow-up (an association like this:

IPI BMI at study follow-up CVD risk factors). BMI is clearly related to CVD risk factors; however, why would differences in IPI lead to differences in BMI at follow-up (20 years later)? Are women with shorter IPIs more likely to have different BMIs 20 years after their pregnancy than women with longer IPIs? I could see why you might want to stratify results by BMI at follow-up if you thought it was an effect modifying variable (not just a mediating variable): i.e., do results for the association between IPI & CVD risk factors vary by maternal BMI at follow-up? (You might imagine that IPI has a stronger effect among women with a higher or lower BMI at follow-up). If you are interested in effect modification, you should stratify by maternal BMI at follow-up rather than just controlling for it.

Otherwise, please explain the reasoning for your decision.

3. Possible bias by time of measurement (before the first pregnancy versus before the second): Do your results differ by whether mothers completed questionnaires before their first pregnancy or before their second? And do these stratified results differ from what you found in the dataset as a whole? Since several of these variables may be different over time (especially pre-pregnancy BMI, smoking during pregnancy, and even being in a single parent household or having a different educational level), women who were measured before their 1st pregnancy may be different than mothers who were measured before their second, & the associations between IPI & CVD may look different due to the timing of measurement. It would be helpful to examine whether the results differ by this variable. In particular, women who were measured before their second pregnancy may be subject to some reverse causality bias: their IPI might actually affect their pre-pregnancy BMI. For instance, women with a short IPI may not have much time to lose weight after their first pregnancy and may thus have a higher pre-pregnancy BMI before their second pregnancy (women with longer IPIs may have more time to lose

weight & lower pre-pregnancy BMIs before their 2nd pregnancy). This is clearly not a problem when the confounders are measured before the 1st pregnancy.

9 (results address research q. or objective): There are large limitations in how confounders were measured (before 1st pregnancy for some women and before 2nd pregnancy in others); in how IPI was defined (by subtracting average gestational age [GA] from each pregnancy rather than using the GA of each individual pregnancy), and regarding the high proportion of missing data. I believe that these large limitations make it very difficult for the results to accurately address the research question/objective.

10 (results presented clearly)- In general, the results were presented clearly, but in Web tables 4 and 5, you have 'inter-birth interval' in the title of the tables rather than 'inter-pregnancy interval.' Also, you did not mention that you adjusted for age in the text (under the "Measurement of confounding factors" section). It was mentioned in the footnotes to the tables but not in the text, so it'd be helpful to change this.

11 (are discussion/conclusions justified)- Due to the large limitations discussed above, I do not think that the conclusions can be justified by the results.

12 (limitations discussed)- I do not think the limitations are adequately discussed. First of all, the method of calculation for IPI is problematic and can lead to non-differential misclassification of exposure. Subtracting 9 months from the interbirth interval rather than the individual's actual GA means that preterm births will have an artificially short IPI, and postterm births will have an artificially long IPI. This would result in a bias toward the null. The authors do not seem to think their method would lead to bias, but their results could be biased toward the null because of this misclassification of exposure. They should at least mention the bias toward the null that can result. The authors also need to spend a bit more time discussing the problems with measuring covariates at different times for certain mothers (e.g., how measuring pre-pregnancy BMI before the start of the second pregnancy could be subject to reverse causality bias). Furthermore, the authors do not believe that associations between IPI & CVD outcomes would differ by women with missing versus non-missing data, but I think that the results could easily differ for women included vs. not included in the study. The extent of missing data is very large in this study. First of all, women with a full reproductive history were quite different than women without a full reproductive history with regard to many covariates. This makes it likely that their IPIs were different (perhaps longer, if they were of higher SES & more educated) and that their CVD profiles were different, too (perhaps more likely to have CVD since they smoke, or conversely, less likely to have CVD since they are highly educated). The results from the women with a full reproductive history are likely not generalizable to all women in the ALSPAC study, and I believe that this should be mentioned. (The authors mention that results likely aren't generalizable to non-white, non-European women, but this lack of generalizability may go further.) Even among the women with full reproductive history, there is a large loss to follow-up, and almost half of these women did not have data on outcomes or covariates. It is unclear if the women with missing data had similar IPIs as women without missing data.

Imputation is a nice attempt to fix the loss to follow-up problem, and it is encouraging that results are similar in the imputed & non-imputed groups. However, there are so many large problems with exposure measurement (discussed above), timing of covariate measurement (discussed above), and lack of generalizability & selection bias that the authors must be very careful about drawing any conclusions. Furthermore, data imputation does not fully solve the missing data problem.

*Interesting research question. Good, high-quality outcome measurement. Nicely written paper.

*I like your suggestions for future research.

*Imputation was a good idea.

*Nice tables.

*Did women who had missing data on outcomes/covariates have similar IPIs as women who had complete data on outcomes/covariates? I did not see this information anywhere. It would be great background information to have as you consider the amount and degree of bias in the study.

*The flow chart is helpful in describing study attrition and selection criteria, but you should fully describe it in words at the beginning of your results section. In particular, please describe in the text that out of ~14,000 recruited women for the ALSPAC study, only ~3500 had full reproductive histories. This is an important number that is mostly only found in tables and in the flow chart; I had to keep scrolling back and forth between the figure and the text to try to figure out how many women with full reproductive histories were included.

*Under the section 'Assessment of Cardiovascular risk factors,' you should indicate that these 11,264 women were not all eligible for the particular study described in this paper (since many of these 11264 women did not have a full reproductive history and were excluded from your analyses). It is clear that the 11264 women who were invited to the ALSPAC clinic follow-up in 2009-2011 were eligible for something (i.e., I assume they were eligible to continue being in the ALSPAC study), but the wording is very confusing because the description makes it sound like all 11264 women are eligible for your particular study on IPI. It is nice to know that 44% of these 11264 women completed outcome assessment, but please reiterate in this section that not all of these women had complete reproductive history.

*Table 4- make "a" and "b" (referring to whether model was unadjusted or adjusted) superscripts

*Do you think there might be effect modification (interaction) by any variables?

*Is there any way to put your results for the imputed & non imputed groups side by side? This would help demonstrate that the results did not differ largely.

VERSION 1 – AUTHOR RESPONSE

Reviewer Name Janet Catov

1. It is not clear at what time age is reported for Table 1. I had thought it was age at the index ALSPAC birth, but that does not seem to be the case given the ages provided in Table 2. The age reported in table 1 is the age at birth of the ALSPAC index child. The average age in this table is different to the average ages reported in table 2 because the first table compares women included in our analyses with those excluded due to missing data on reproductive history, and the second table reports only on women included in our analyses, stratified according to inter pregnancy interval. In response to this comment we have changed the row in table 1 from “Age” to “Age at birth of the ALSPAC index child”

2. It is not clear why the mean values for the entire measured cohort (~4000) are reported in table 3 when they are not compared to the subcohort analyzed. It perhaps would have been more informative to see the mean values summarized by interval timing, as for table 2

We agree with the reviewer and have now changed Table 3 to include only women included in our analysis, and have stratified the data according to inter pregnancy interval. The following text in the manuscript has been updated:

“Table 3 presents the distribution of the outcome variables for all those women who attended the follow-up assessment in 2009-11 and who had an IPI calculated for this analysis.” (page 12)

3. Tables 4 &5 are a bit hard to follow. IBI is not defined. It appears that the models noted as ‘a’ are unadjusted, and ‘b’ are adjusted; but then I am not certain what the numbers refer to (1a, 2b)

Thank you for this comment. Inter pregnancy interval is defined in the columns of the table; ≤ 15 months and >27 months are each compared with the reference category of 16-27 months. A typo lead to ‘IBI’ being included in an incorrect column title, we have now corrected this. The numbers 1a, 2b etc refer to the statistical analysis models, which are detailed in the footnotes. We have now made this clearer by adding a column title called ‘Analysis Model Number’

Reviewer Name Cassandra Gibbs Pickens, MPH

2 (abstract)-You need to mention attrition/missing data in the abstract since it is such a large limitation of this study. You should mention the % of women in the ALSPAC Study with full reproductive history (approximately 25%). You should also mention the % of women with full reproductive history (& who are eligible due to having exactly 2 live births) who have non-missing data (approximately 56%). You should also mention how you determined interpregnancy interval (interbirth interval minus the average GA, 9 months) in the abstract.

The results section of the abstract has now been changed to reflect the reviewer’s comments:
“Twenty five percent (n=3451) of ALSPAC mothers had provided sufficient data to determine full reproductive history - of these, 1477 had two live births, with 54% mothers having non-missing data on all variables required for our analyses. A total of 1268 mothers with IPI (inter-birth interval minus 9 months gestation) had cardiovascular disease risk factors measured/imputed at mean age 48 years..”

1. What about control for breastfeeding duration? Breastfeeding may affect IPI (through lactation-induced amenorrhea) and may also decrease the risk of adverse CVD outcomes later in life and could be an important confounder to include in your adjusted models. I suppose you only have information on breastfeeding for women who filled out questionnaires for their second pregnancy (not their first), but it would be a good confounder to include if you ran a model among these women only. If you don’t have this information, listing it as a potential limitation would be a good idea.

We agree with the reviewer that breast feeding duration maybe an additional confounder. In order to

evaluate the role of breastfeeding duration as a potential confounder of our associations, we included breastfeeding duration (Never, ≤ 1 month, 2-4 months, 5-6 months and >6 months) in our fully adjusted model in women with complete data ($n=785$), and compared the results in these women with and without adjustment for breastfeeding duration (see separate supplementary table for results with and without adjustment for breastfeeding). The results were almost identical with and without adjustment for breastfeeding, demonstrating that duration of breastfeeding is not an important confounder of our associations. We have therefore not altered the main paper.

2. Why do you include BMI at follow-up as a mediating variable for the IPI CVD risk factor association? Controlling for maternal BMI at follow-up indicates that some of the IPI CVD risk factor association is mediated through maternal BMI at follow-up (an association like this: IPI BMI at study follow-up CVD risk factors). BMI is clearly related to CVD risk factors; however, why would differences in IPI lead to differences in BMI at follow-up (20 years later)? Are women with shorter IPIs more likely to have different BMIs 20 years after their pregnancy than women with longer IPIs? I could see why you might want to stratify results by BMI at follow-up if you thought it was an effect modifying variable (not just a mediating variable): i.e., do results for the association between IPI & CVD risk factors vary by maternal BMI at follow-up? (You might imagine that IPI has a stronger effect among women with a higher or lower BMI at follow-up). If you are interested in effect modification, you should stratify by maternal BMI at follow-up rather than just controlling for it. Otherwise, please explain the reasoning for your decision.

We believe BMI at follow-up is an important potential mediator of the IPI – cardiovascular health associations. One hypothesis is that shorter IPI could lead to increased weight retention after pregnancy and/or greater gestational weight gain in the second pregnancy. We wished to examine the question of whether any influence of IPI on later cardiovascular health was entirely explained by BMI, or whether any independent associations existed.

3. Possible bias by time of measurement (before the first pregnancy versus before the second): Do your results differ by whether mothers completed questionnaires before their first pregnancy or before their second? And do these stratified results differ from what you found in the dataset as a whole? Since several of these variables may be different over time (especially pre-pregnancy BMI, smoking during pregnancy, and even being in a single parent household or having a different educational level), women who were measured before their 1st pregnancy may be different than mothers who were measured before their second, & the associations between IPI & CVD may look different due to the timing of measurement. It would be helpful to examine whether the results differ by this variable. In particular, women who were measured before their second pregnancy may be subject to some reverse causality bias: their IPI might actually affect their pre-pregnancy BMI. For instance, women with a short IPI may not have much time to lose weight after their first pregnancy and may thus have a higher pre-pregnancy BMI before their second pregnancy (women with longer IPIs may have more time to lose weight & lower pre-pregnancy BMIs before their 2nd pregnancy). This is clearly not a problem when the confounders are measured before the 1st pregnancy.

In response to this reviewer's comment we conducted an additional set of analysis (complete case) to compare the results from the main paper with the results stratified by whether the ALSPAC pregnancy was the first or second pregnancy for this study. There were some slight variations to the effect estimates when stratified, however, the confidence intervals of the stratum specific effect estimates overlapped with each other and with the main analysis estimates when first and second pregnancy data were combined. The small differences we observed were also not consistent across the outcomes, suggesting that these differences are likely to be chance findings, and therefore we have concluded that the results do not differ between the stratified and the main results. This additional analysis has created 6 additional tables, which have not been included as supplementary data in order to maintain the clarity and readability of the paper, but we have indicated in the main paper that this sensitivity analysis has been performed and that the results are available from authors on

request. The following sentences have been included into the manuscript:

“In order to assess whether our results were biased as a result of some confounders being measured at the first pregnancy of the IPI for some women and the second for others, we stratified our results by whether the ALSPAC child was the first or second pregnancy of the interval. “ (page 9)

And

“The results were also similar when the women were stratified according to whether the ALSPAC child was the first or second pregnancy (results not shown – available from authors on request)
“(page 15)

9 (results address research q. or objective): There are large limitations in how confounders were measured (before 1st pregnancy for some women and before 2nd pregnancy in others); in how IPI was defined (by subtracting average gestational age [GA] from each pregnancy rather than using the GA of each individual pregnancy), and regarding the high proportion of missing data. I believe that these large limitations make it very difficult for the results to accurately address the research question/objective.

We thank the reviewer for their comment. We agree that these factors limit the strength of conclusion that can be reached from these analyses. However, we feel that we have carefully documented and discussed these limitations in the manuscript so that readers are aware of the study design and the assumptions underlying our analysis. Furthermore, we have carried out several sensitivity analyses to test these assumptions as far as possible; in response to the reviewer’s suggestion we have also added additional sensitivity analyses showing that the results are similar when stratifying according to whether the ALSPAC pregnancy was the woman’s first or second, and have showed that the results are similar for these two subsets of women. Finally, we feel it is important to remember that very few studies have prospectively collected data on pregnancy exposures as well as data on cardiometabolic health in mid-life, particularly with a sufficient sample size to be able to restrict analysis to women with two live births in order to remove confounding by parity. Therefore despite these limitations, we feel the study makes a valuable contribution to the literature in this field, and will hopefully spark discussion and further research on this topic in other cohorts.

10 (results presented clearly)- In general, the results were presented clearly, but in Web tables 4 and 5, you have 'inter-birth interval' in the title of the tables rather than 'inter-pregnancy interval.'

Thank you for pointing out this error, we have corrected this now to read “inter-pregnancy interval” in the web tables 4 and 5.

Also, you did not mention that you adjusted for age in the text (under the "Measurement of confounding factors" section). It was mentioned in the footnotes to the tables but not in the text, so it'd be helpful to change this.

Under the “measurement of confounding factors” in the methods section we have added a sentence at the end of the paragraph: “Participant’s age at clinic attendance was included as a confounding factor.”

11 (are discussion/conclusions justified)- Due to the large limitations discussed above, I do not think that the conclusions can be justified by the results.

As discussed in response to the reviewer’s point 9, we feel that we have discussed the limitations of the study in detail, and have carried out sensitivity analyses to test, as far as is possible, the assumptions of our analysis. We have also been cautious in our conclusion statement at the end of the discussion, and in response to the reviewer’s comment we have added further comment to this as follows (new text underlined): “In conclusion, our results do not support an association between IPI and cardiovascular risk factors, though our findings must be interpreted in light of the large losses to follow-up and limitations in our measurement of IPI. Further studies in other populations with more

detailed data on gestational age at delivery of all pregnancies, in different settings such as low income countries where the social patterning of IPI may differ and in older women with greater inter-individual variability in cardiovascular risk factors would provide a more comprehensive understanding of these associations.” Furthermore, we reiterate the dearth of studies on this topic, and the strengths of our analyses, including the large sample size, ability to restrict to women with two live births in order to control for confounding by parity, and detailed measurements of cardiometabolic health in mid-life.

12 (limitations discussed)- I do not think the limitations are adequately discussed. First of all, the method of calculation for IPI is problematic and can lead to non-differential misclassification of exposure. Subtracting 9 months from the interbirth interval rather than the individual's actual GA means that preterm births will have an artificially short IPI, and postterm births will have an artificially long IPI. This would result in a bias toward the null. The authors do not seem to think their method would lead to bias, but their results could be biased toward the null because of this misclassification of exposure. They should at least mention the bias toward the null that can result.

We acknowledge that it is possible that we may have introduced a misclassification bias by calculating IPI in this manner. We do not think that our results would be greatly biased by this (as we mention on page 20 of our discussion). However, as the reviewer mentions there is a possibility that our results could be attenuated towards the null. In response to this we have added in the following sentence into the limitations section of the discussion: “It is possible that by calculating our IPI in this way we have attenuated our results towards the null.” (page 19) We have also added to the conclusion statement that the method of calculating IPI is a potential source of bias in our study, as detailed in our response to point 11 above.

The authors also need to spend a bit more time discussing the problems with measuring covariates at different times for certain mothers (e.g., how measuring pre-pregnancy BMI before the start of the second pregnancy could be subject to reverse causality bias).

We agree that it is potentially problematic that the confounders are measured for the second pregnancy for some women and for the first pregnancy for others. However, this cannot cause reverse causality bias – in which the outcome (cardiometabolic health) causes the exposure (IPI). Given that adjustment for pre-pregnancy BMI had little impact on the results (including when restricting to analysis of women where confounders were measured during their first pregnancy), we do not believe that this has caused bias in our results. However, in response to the reviewer's suggestion in point 3 above, we have now carried out a further sensitivity analysis and have verified that our results are similar when stratifying according to whether the ALSPAC child was the first or second pregnancy; see detailed response to point 3

Furthermore, the authors do not believe that associations between IPI & CVD outcomes would differ by women with missing versus non-missing data, but I think that the results could easily differ for women included vs. not included in the study. The extent of missing data is very large in this study. First of all, women with a full reproductive history were quite different than women without a full reproductive history with regard to many covariates. This makes it likely that their IPIs were different (perhaps longer, if they were of higher SES & more educated) and that their CVD profiles were different, too (perhaps more likely to have CVD since they smoke, or conversely, less likely to have CVD since they are highly educated). The results from the women with a full reproductive history are likely not generalizable to all women in the ALSPAC study, and I believe that this should be mentioned. (The authors mention that results likely aren't generalizable to non-white, non-European women, but this lack of generalizability may go further.) Even among the women with full reproductive history, there is a large loss to follow-up, and almost half of these women did not have data on outcomes or covariates. It is unclear if the women with missing data had similar IPIs as women without missing data. Imputation is a nice attempt to fix the loss to follow-up problem, and it is encouraging that results are similar in the imputed & non-imputed groups. However, there are so many large problems with exposure measurement (discussed above), timing of covariate measurement (discussed above), and lack of generalizability & selection bias that the authors must

be very careful about drawing any conclusions. Furthermore, data imputation does not fully solve the missing data problem.

We acknowledge that missing data is a limitation of this study. We have made attempts to limit the amount of missing data by using multiple imputation, but we did not impute for those women who we did not have the exposure data for IPI since we do not have sufficient data on which to base the imputation of IPI. As pointed out by the reviewer, the women included in our analysis tend to be of higher socioeconomic position than the full ALSPAC cohort. The reviewer states that our results may not be generalizable to women in a lower social class; we agree that our population differ from the general population, however lack of generalisability of a study population does not always confer selection bias. If a causal aetiological relationship between IPI and cardiometabolic health existed, we would expect to see this even in a relatively affluent population. Studies from registry data in Scandinavia have provided several examples of this phenomenon[1-4]. Although we cannot rule out selection bias, given the lack of association between IPI and cardiometabolic health in our relatively affluent population, any observed association within a lower socioeconomic population may well be the result of confounding rather than a true causal effect. We do, however, recognise that we cannot prove whether our results are affected by selection bias. Therefore, in response to the reviewer's comment we have included into the limitations section of the discussion the following sentences: " This means that the women included in our analysis are not a representative sample of the full ALSPAC cohort; some studies suggest that lack of generalisability does not necessarily result in selection bias[1-4], but we cannot be certain of this from the current analysis and replication of our results in other studies with different distributions of socio-demographic variables would be beneficial. " (page 19)

And

"A further limitation of this study is these findings are only generalizable to a largely white European population with a higher socioeconomic status." (page 219)

We have also added the following sentence to the conclusion: "In conclusion, our results do not support an association between IPI and cardiovascular risk factors, though our findings must be interpreted in light of the large losses to follow-up and limitations in our measurement of IPI." (page 19-20)

REFERENCES

- 1 A. J. Sogaard, R. Selmer, E. Bjertness, et al., The Oslo Health Study: The impact of self-selection in a large, population-based survey. *International journal for equity in health* 2004;3:3
- 2 A. J. Van Loon, M. Tjhuis, H. S. Picavet, et al., Survey non-response in the Netherlands: effects on prevalence estimates and associations. *Ann Epidemiol* 2003;13:105-10
- 3 E. A. Nohr, M. Frydenberg, T. B. Henriksen, et al., Does low participation in cohort studies induce bias? *Epidemiology* 2006;17:413-8
- 4 R. M. Nilsen, S. E. Vollset, H. K. Gjessing, et al., Self-selection and bias in a large prospective pregnancy cohort in Norway. *Paediatric and perinatal epidemiology* 2009;23:597-608
- 5 A. Fraser, C. Macdonald-Wallis, K. Tilling, et al., Cohort Profile: The Avon Longitudinal Study of Parents and Children: ALSPAC mothers cohort. *Int J Epidemiol* 2012;97-110

*Did women who had missing data on outcomes/covariates have similar IPIs as women who had complete data on outcomes/covariates? I did not see this information anywhere. It would be great background information to have as you consider the amount and degree of bias in the study. In response to this comment we have included an additional web table (table 1) which compares the IPI of women with complete data vs. women with missing data. Women with complete data had similar IPIs to those with missing data. We have also included the following sentence to the results section: "Women included in the complete case analysis versus women with missing data on covariates and/or outcome variables had similar IPIs (web table 1)." (page 9)

*The flow chart is helpful in describing study attrition and selection criteria, but you should fully describe it in words at the beginning of your results section. In particular, please describe in the text that out of ~14,000 recruited women for the ALSPAC study, only ~3500 had full reproductive histories. This is an important number that is mostly only found in tables and in the flow chart; I had to keep scrolling back and forth between the figure and the text to try to figure out how many women with full reproductive histories were included.

We have added in further information to the first sentence of the results section: "A total of 3451 women out of 13,713 recruited women who had a live birth had sufficient data to describe a full reproductive history" (page 9)

*Under the section 'Assessment of Cardiovascular risk factors,' you should indicate that these 11,264 women were not all eligible for the particular study described in this paper (since many of these 11264 women did not have a full reproductive history and were excluded from your analyses). It is clear that the 11264 women who were invited to the ALSPAC clinic follow-up in 2009-2011 were eligible for something (i.e., I assume they were eligible to continue being in the ALSPAC study), but the wording is very confusing because the description makes it sound like all 11264 women are eligible for your particular study on IPI. It is nice to know that 44% of these 11264 women completed outcome assessment, but please reiterate in this section that not all of these women had complete reproductive history.

In response to this comment, we have now deleted the word "eligible" from the first sentence under the section 'Assessment of cardiovascular risk factors' to avoid confusion. The sentence now reads: "Between 2009 and 2011 ALSPAC mothers (N=11, 264 women) were invited to a research clinic assessment at which a range of cardiovascular outcomes were assessed; this clinic took place between 1.6 and 20.3 years (median 18) since the second birth defining the end of the the IPI." (page 6)

We have also added the following sentence to the end of the first paragraph in this section: "Not all of these women were eligible for inclusion in our analyses as not all of them had full reproductive histories recorded." (Page 6)

We have also updated the paragraph at the end of "Assessment of cardiovascular risk factors" section. We have changed the number of women who attended clinic from 5005 to 4834. The previous 5005 women reported to have attended clinic did not exclude women who had more than one pregnancy enrolled in ALSPAC nor did it exclude women with no live births or unknown outcomes. This was an oversight. The corresponding percentage has also changed to reflect this and a reference added to the cohort paper published elsewhere. We have also included reference to the number of these 4834 women who had a full reproductive history reported (n=954). The paragraph now reads:

"A total of 4834 women attended clinic (43% response)[5]. Women attending clinic had a mean age of 48 years Women pregnant at the time of clinic assessment were excluded from our analyses (n=7). Only 954 women attending clinic and not pregnant at the time of clinic assessment, had full reproductive histories."(page 7)

*Table 4- make "a" and "b" (referring to whether model was unadjusted or adjusted) superscripts

The labelling in table 4 has now been changed so that the superscripts have been reinstated – therefore removing the numbers 1a and 2b. These now read 1a and 2b, which relate to the footnotes.

*Do you think there might be effect modification (interaction) by any variables?

We have no a priori hypotheses about effect modification, and as such we have not carried out statistical tests for interaction given the risk of false positive results.

*Is there any way to put your results for the imputed & non imputed groups side by side? This would help demonstrate that the results did not differ largely.

There are many outcomes in this analysis and as a result it was difficult to present these in comprehensible tables. We tried to present the imputed and non-imputed results side by side, but this made the tables difficult to read and they were too big to fit onto one side of A4. Therefore we have left the tables as they are.